A novel deep neural network structure for software fault prediction

Modanlou Jouybari Mehrasa 1
Tajary Alireza 1 tajary@gmail.com
Fateh Mansoor 1
Abolghasemi Vahid 2 v.abolghasemi@essex.ac.uk
1 Faculty of Computer Engineering, Shahrood University of Technology , Shahrood , Iran
2 School of Computer Science and Electronic Engineering, University of Essex , Colchester , United Kingdom
Akleylek Sedat
Electronic publication date: 2024 Oct 17
Publication date: 2024
Volume: 10
Electronic Location ID: e2270
Received 2024 Mar 1; Accepted 2024 Jul 28
Copyright: © 2024 Modanlou Jouybari et al.
Copyright year: 2024
Copyright holder: Modanlou Jouybari et al.
License: This is an open access article distributed under the terms of the Creative Commons Attribution License, which permits unrestricted use, distribution, reproduction and adaptation in any medium and for any purpose provided that it is properly attributed. For attribution, the original author(s), title, publication source (PeerJ Computer Science) and either DOI or URL of the article must be cited.
License URL: https://creativecommons.org/licenses/by/4.0/

Keywords: Deep neural network, Software fault prediction, BugHunter dataset, Machine learning

Funding: The authors received no funding for this work.

==============================
Software fault prediction is crucial to compute the potential occurrence of faults within the software components, before code testing or execution. Machine learning, especially deep learning, has been applied to predict faults, but both encounter challenges such as insufficient accuracy, imbalanced data, overfitting, and complex structure. Moreover, deep learning yields superior predictions when working with large datasets. The most common fault dataset is NASA MDP from the PROMISE repository. The BugHunter dataset, however, offers a larger number of instances compare to other fault datasets, leaving a gap in the literature for exploring the application of machine learning and deep learning. In this study, we present a novel structure of deep neural network (DNN), which utilizes the convolutional layers to extract valuable knowledge from the BugHunter data projects. Our proposed model addresses class imbalance and overfitting issues while accurately predicting fault-prone methods. To evaluate the effectiveness of our predictive model, we conduct extensive empirical studies comparing it with seven traditional machine learning, three ensemble learning, and three state-of-the-art deep learning baseline models. Our findings revealed that the proposed DNN structure significantly improved the average F1-score in 15 projects of the BugHunter datasets by 20.01%, indicating that DNN is a practical approach for predicting faulty methods. Leveraging these results could lead to the preservation of software development resources and the production of more reliable software.

Introduction

The development of software entails intricate planning and substantial investments of time and resources. In software systems, the presence of latent and unforeseen faults represents an inevitable challenge (Khan & Nadeem, 2023). A software fault refers to a mismatch between the expected and actual output, which ultimately fails the final product (Akimova et al., 2021). Testing is a vital activity that ensures the production of fault-free and high-quality software. However, it is a challenging, near-impossible, and costly undertaking for testers to uncover all faults within a software system (Khan & Nadeem, 2023; Aziz, Khan & Nadeem, 2021). Identifying more fault-prone modules can facilitate the process of testing, since 80% of faults are typically found in only 20% of the modules (Muhammad, Nadeem & Sindhu, 2021). Software fault prediction (SFP) emerges as a valuable approach to detect fault-prone modules, prior to code testing or execution. SFP enables targeted resource allocation, maintenance cost reduction, and assuring the quality of the software upon deployment (Zhu et al., 2021).

Artificial intelligence (AI) empowers machines to mimic human behaviors, while machine learning (ML), a subset of AI, enables machines to learn from experience (Ertel, 2018). Numerous studies have extensively explored and applied diverse ML models such as multi-layer perceptron (MLP) (Rathore & Kumar, 2017), K-nearest neighbors (KNN) (Pandey, Mishra & Tripathi, 2021), naive Bayes (NB) (Arar & Ayan, 2017), Decision Tree (DT) (Khan et al., 2020), logistic regression (LR) (Chen et al., 2018), Random Forest (RF) (Matloob et al., 2021), and support vector machine (SVM) (Wang et al., 2021) to predict faults. However, these models can exhibit variation results in SFP. Deep learning (DL) is a subset of ML that has demonstrated effectiveness in various domains. Through DL, DNNs can leverage deep architectures that consist of multiple layers of artificial neurons. This characteristic enables DNNs to acquire intricate patterns and representations from raw input data by employing various complex functions and non-linear transformations (Jia et al., 2016). Two prominent types of DNNs are convolutional neural networks (CNNs) and long short-term memory networks (LSTMs). These networks are widely employed in complex problems, including object detection, pattern recognition, speech recognition, and natural language processing (Voulodimos et al., 2018; Qiao et al., 2020). The most commonly used dataset for SFP is the NASA MDP from PROMISE repository (Malhotra, 2015). To the best of our knowledge, the BugHunter dataset (Ferenc et al., 2020) contains a larger number of data instances compared to other existing fault datasets, making it a potential candidate to achieve higher performance with DNNs.

The BugHunter dataset introduces the challenge of class imbalance, which researchers have attempted to address using the under-sampling technique (Ferenc et al., 2020). However, this technique may discard valuable instances from the majority class (Feng et al., 2021). To overcome this, we explore the application of the over-sampling technique to preserve the data instances and handle the class imbalance. Additionally, the state-of-the-art models exhibit challenges related to overfitting and inaccurate prediction performance when applied to the BugHunter dataset. This study focuses on addressing the aforementioned gaps by examining the performance and efficiency of a range of ML and DL models on 15 projects of the BugHunter dataset with the purpose of mitigating overfitting and improve fault prediction performance. In summary, this research provides the following contributions: Addressing class imbalance: To handle the class imbalance in the BugHunter dataset, we use the oversampling method to enhance the accuracy of SFP for both fault classes.

Reducing overfitting and increasing accuracy: We propose a novel DNN structure utilizing CNNs, a DL architecture to extract and learn valuable knowledge from the BugHunter data projects, reduce overfitting, and accurately predict which software methods are faulty or non-faulty.

Transforming the dataset into three-dimensional representation: In order to utilize the windowing technique of two-dimensional convolutional layers, we converted the dataset into a fresh three-dimensional representation. This adjustment enables a greater volume of data to be accessible for each window, enhancing the model’s ability to extract information from the dataset.

Building and evaluating different DNN structures: We investigate the influence of various DNN architectures with unique structures and conditions. This is to improve the model’s ability to detect faulty methods. We thoroughly study the effects of modifications such as altering filter sizes, incorporating max pooling layer, employing two-dimensional CNNs, etc.

Improving the DNN performance: We introduce the optimal DNN model by adjusting four hyper-parameters (dropout rate, number of epochs, batch size, and learning rate) for each of the 15 Java projects. We also explore the influence of various kernel sizes on the performance and efficiency of the proposed model.

Assessing our DNN model performance against various SFP baseline methods: Our comprehensive empirical investigations involve evaluating the efficacy of our proposed model against seven conventional ML baseline models, three renowned ensemble learning baseline models, three cutting-edge DL baseline models, and the current performance metrics on BugHunter datasets achieved by 11 traditional models.

Comparing the efficiency of different SFP classifiers: We present the efficiency of various SFP models employed in this study, considering their training and testing times, and identify the most optimal model among them.

Related works

SFP has attracted considerable interest from researchers seeking to enhance the reliability and quality of software systems over time. A common approach to accomplish SFP involves categorizing software modules/classes into faulty and non-faulty groups (Batool & Khan, 2022; Malhotra, 2015). Numerous studies have delved into a wide range of ML and, especially, DL classifiers for efficient SFP, utilizing various datasets and performance measures. This section provides a review of these models and a discussion of existing challenges regarding SFP. 1) Traditional ML techniques:

ML techniques have gained significant attention in SFP for their ability to analyze various software metrics and predict the occurrence of software faults. Rathore & Kumar (2019) observed that a majority of recent studies employed supervised learning techniques including DT, SVM, MLP and ensemble methods like RF. These were followed by statistical techniques encompassing LR, KNN, and NB for building the SFP model. The simplicity of these techniques and the absence of the need for complex parameter optimization made them particularly appealing. In a systematic literature review conducted by Malhotra (2015), the performance of ML techniques in SFP was thoroughly analyzed and assessed. The results prove the predictive capabilities of ML techniques in classifying modules/classes as either fault-prone or non-fault-prone. The review identified C4.5, NB, MLP, SVM, and RF as the most frequently employed ML techniques for SFP. Ferenc et al. (2020) utilized the Weka library to construct 11 traditional ML models, including NB, Naive Bayes Multinomial, Logistic, SGD, Simple Logistic, Voted Perceptron, Decision Table, OneR, J48 (C4.5), RF, and Random Tree. Their objective was to evaluate the efficacy of their novel dataset, BugHunter, in predicting software faults, and they achieved F-measure values up to 0.75. Pandey, Mishra & Tripathi (2021) investigated the competence of ML techniques in SFP. They identified seven prominent categories of ML techniques widely employed in SFP, including Bayesian learners, DTs, Evolutionary Algorithms, Ensemble Learners, Neural Networks, SVMs, Rule-Based Learning, and Miscellaneous approaches. Among these categories, RF, SVM, NB, C4.5, LR, and MLP were the most commonly used techniques for SFP. Cynthia, Roy & Mondal (2022) introduced a feature transformation technique for feature extraction. This technique involved identifying a weighted transformation of these features through a genetic algorithm to optimally distinguish faults from non-faults in a reduced-dimensional space, and subsequently employing the transformed dataset for training ML models. The effectiveness of this approach was assessed by training RF, KNN, LR, and NB models on both the original and transformed datasets across seven BugHunter data projects at the class level. 2) DL-based techniques:

Deep learning is a subfield of ML that is preferable in SFP when dealing with large datasets (over 10k instances) and yields superior predictions (Pandey, Mishra & Tripathi, 2021). Farid et al. (2021) utilized two main DL models, CNN and bidirectional long short-term memory (Bi-LSTM), to introduce the CBIL framework. The experimental results show that CBIL is successfully predicts software faults, leading to a 30% improvement over baseline models and a 25% enhancement over CNN in terms of average F-measure. In another work, Zain et al. (2022) proposed a one-dimensional convolutional neural network (1D-CNN), a DL architecture based on the NASA datasets, for predicting the fault-proneness of code units. The results revealed that the proposed 1D-CNN classification model outperformed other CNN and traditional ML models regarding accuracy and f-measure. Based on the findings, the 1D-CNN is an effective SFP model. Qiao et al. (2020) presented a DL-based model to predict the number of faults in software modules. The model utilized software metrics derived from two real-world datasets: the Medical Imaging System dataset and the KC2 dataset from NASA PROMISE. The evaluation results reveal that the DL-based model is accurate and significantly outperforms the state-of-the-art models. 3) Discussion:

Study of the relevant literature reveals that traditional ML techniques cannot effectively capture both the syntax and various levels of semantics present within the source code, which are essential for constructing accurate prediction models. On the contrary, DL algorithms demonstrate the capability to extract semantic information from software metrics, enabling them to make accurate predictions (Omri & Sinz, 2020). In particular, DL is effective when researchers consider larger datasets (Batool & Khan, 2022).

Most of the SFP datasets exhibit imbalanced characteristics (Pandey & Kumar, 2023; Japkowicz & Stephen, 2002; Tantithamthavorn, Hassan & Matsumoto, 2018), with a majority number of non-faulty instances and a minority number of faulty instances, resulting in a skewed class distribution and biased outcomes (Giray et al., 2023). To tackle this problem, commonly employed techniques involve utilizing over-sampling and under-sampling methods (Song, Guo & Shepperd, 2018; Tantithamthavorn, Hassan & Matsumoto, 2018). Overfitting is another concern that must be addressed in SFP (Pandey, Mishra & Tripathi, 2021), as ML models often demonstrate superior performance on the train set but struggle to generalize effectively on the test set (Santos & Papa, 2022). To overcome this challenge, it is recommended to employ ensemble learning and k-fold cross-validation techniques (Huda et al., 2018; Pandey, Mishra & Tripathi, 2021). Furthermore, regularization methods were utilized in neural networks, providing a means to alleviate model overfitting, as employed in Stacked Auto-Encoder architectures (Omri & Sinz, 2020; Tong, Liu & Wang, 2018) and CNNs (Santos & Papa, 2022). Moreover, the current SFP models’ performance relies on two factors: the specific system and the fine-tuning of the model hyper-parameters (Pandey, Mishra & Tripathi, 2021). Hyper-parameters tuning is the crucial determinant of the model’s optimal performance. However, the extent of research conducted on the tuning of model hyper-parameters is limited.

The proposed approach

Figure 1 depicts the overall framework of this study which designs and develops various SFP classifiers. It includes two primary phases: data preprocessing and model construction. The first phase involves collecting 15 Java projects from the BugHunter dataset and applying preprocessing techniques on them. The second phase focuses on constructing and evaluating various predictive models to increase the SFP performance.

Figure 1 Overall framework of the proposed approach.

Phase 1) preprocessing

Data preprocessing is crucial for ensuring high-quality data and preventing misleading outcomes (García, Luengo & Herrera, 2015). In the preprocessing phase, we conducted the following steps.

The BugHunter dataset consists of both nominal and numerical features. Existence of nominal features in the dataset can present difficulties, notably when the chosen algorithm cannot handle them appropriately (García, Luengo & Herrera, 2015). To facilitate the implementation of our algorithm, we employed Label Encoder (Gupta & Asha, 2020) to map the nominal features, ‘Hash’ and ‘LongName’, into numerical representations. Given that eliminating the irrelevant features can enhance the understanding of the extracted patterns and facilitate a faster learning process (García, Luengo & Herrera, 2015), for each project, we removed numerical features with no difference between their maximum and minimum values.

In machine learning, it is typical to utilize cross-validation for assessing the generalizability of the predictive models (Salazar et al., 2022). To accomplish this, we divided the data into training and testing sets using StratifiedKFold cross-validation from the sikit-learn package, employing ten folds. Traditional ML models exhibit superior performance on scaled data that does not include significant discrepancies in feature values. To ensure a fair comparison between different models, we employed the MinMaxScaler from the scikit-learn package (Pedregosa et al., 2011) to scale the software metrics into the interval [0,1].

Next, we address the class imbalance. This issue occurs when the frequency of faulty methods/files/classes is much lower than that of non-faulty methods/files/classes. Such an imbalance can lead to a biased model, making it impractical for real-world applications (Feng et al., 2021). To address this issue, two commonly employed techniques are oversampling and under-sampling. Oversampling involves replicating instances within the minority class, while under-sampling entails eliminating instances from the majority class (Farid et al., 2021). In this study, the RandomOverSampler method from the imblearn library was utilized to apply the oversampling technique to preserve the data instances. Table 1 presents the dataset description before and after preprocessing, the number of instances, and the imbalance ratio. Since the imbalance problem has been severe, in addition to oversampling, we will also employ the class weight method to address this issue.

Table 1 The description of the BugHunter dataset before and after preprocessing.

Name of the project in the BugHunter	Total # of instances before prep.	# of faulty instances before prep.	# of non-faulty instances before prep.	Faulty ratio (%)	Imb. ratio	# of software metrics after prep.	Total # of instances after prep.	# of faulty instances after prep.	# of non- faulty instances after prep.	
ceylon-ide-eclipse	2,087	508	1,579	24.34	3.11	58	2,972	1,393	1,579	
BroadleafCommerce	4,709	1,025	3,684	21.77	3.59	61	6,824	3,140	3,684	
hazelcast	32,973	12,093	20,880	36.68	1.73	61	39,923	19,043	20,880	
elasticsearch	35,862	11,950	23,912	33.32	2	62	45,497	21,585	23,912	
MapDB	1,456	480	976	32.97	2.03	59	1,842	866	976	
netty	11,171	2,434	8,737	21.79	3.59	59	16,207	7,470	8,737	
orientdb	9,445	2,589	6,856	27.41	2.65	61	12,911	6,055	6,856	
neo4j	7,030	1,841	5,189	26.19	2.82	59	9,704	4,515	5,189	
titan	785	168	617	21.4	3.67	61	1,147	530	617	
mcMMO	1,184	411	773	34.71	1.88	55	1,493	720	773	
Android-Universal-Image-Loader	325	103	222	31.69	2.16	51	415	193	222	
antlr4	840	102	738	12.14	7.24	56	1,350	612	738	
junit	462	87	375	18.83	4.31	57	695	320	375	
mct	105	25	80	23.81	3.2	53	143	63	80	
oryx	810	77	733	9.51	9.52	55	1,350	617	733	

Phase 2) building the model

A DL architecture called DNN is proposed to capture intricate patterns between faulty and non-faulty methods. The proposed DNN comprises convolutional, pooling, and fully connected layers, which together form the base structure of the network.

Pooling layers (Singh, Raj & Namboodiri, 2020) facilitate dimensionality reduction by consolidating information from neighboring neurons within a confined spatial area, thereby decreasing the input size. A prevalent technique, such as max-pooling, selects the highest value within each pooling region. This method aids in downsampling the feature maps and retaining crucial features by preserving the maximum values. A fully connected layer (Qiao et al., 2020) establishes connections between each neuron in one layer and each neuron in the next layer. This layer plays a critical role in facilitating comprehensive interactions between neurons across preceding and subsequent layers, enabling the extraction of high-level features. The structure of the proposed DNN is illustrated in Fig. 2. In the following section, a brief explanation on each layer is provided.

Figure 2 The structure of the proposed DNN model.

The first convolutional layer: as the input layer, the initial convolutional layer receives a set of n software metrics (independent variables/features). The selection of n is determined based on the count of software metrics for each project, as outlined in Table 1. For example, considering the ceylon-ide-eclipse project, the network’s input was configured to include 58 software metrics. This layer processes the input sequence and translates the outcomes onto 64 feature maps using the rectified linear unit (ReLU) activation function and the kernel size three. The convolved features generated by this layer contain more informative knowledge than the original input features.

The second convolutional layer: Building upon the 64 feature maps generated by the first convolutional layer, the second layer conducts the same task utilizing the ReLU activation and the kernel size three, aiming to extract more informative features and complex patterns.

Flatten layer: Following the second convolutional layer, the extracted feature maps are flattened into a single long vector, which can then be fed into the decoding process.

Fully connected layer: Comprising two dense layers and one dropout layer positioned between them, utilizing a dropout rate 0.2. This layer plays a crucial role in interpreting each vector within the output sequence and classifying methods as faulty or non-faulty. The first dense layer utilizes the ReLU activation function, while the second one employs the sigmoid. The dropout layer is incorporated to prevent overfitting by temporarily dropping connections between nodes (Srivastava et al., 2014). The network utilized Adam optimization with a 0.01 learning rate and binary cross-entropy loss function, a common approach when dealing with classification problems.

We constructed 17 different DNN classifiers with distinct architectures to explore the influence of diverse structures on the DNN-1 performance. These models were denoted as DNN-2, DNN-3, DNN-4, DNN-5, DNN-6, DNN-7, DNN-8, DNN-9, DNN-10, DNN-11, DNN-12, DNN-13, DNN-14, DNN-15, DNN-16, and DNN-17. The key differences among these models are summarized in Tables 2 and 3. Changes such as filters size for each convolutional, adding/removing max pooling layers and dropout layers are applied. In particular, DNN-5 includes the imbalanced class weights during the learning process of the balanced classes to investigate the effect of assigning higher weights to the minority class. This approach enables the model to prioritize these instances during training, reducing bias towards the majority class and giving them greater importance. To achieve this, we utilized the class weight technique using the Keras library. For each project, the imbalanced weights were calculated first using Eqs. (1) and (2), and then applied during the training of the balanced classes.

Table 2 The proposed model with different structures.

	DNN-1	DNN-2	DNN-3	DNN-4	DNN-5	DNN-6	DNN-7	
#Convolutional layer	2	2	2	2	2	3	2	
Convolutional layer dim	1D	1D	1D	1D	1D	1D	2D	
#Filters	32,32	64,64	32,32	32,32	32,32	32,32,15	32,32	
Kernel size	3,3	3,3	3,3	3,3	3,3	3,3,3	(3,3), (3,3)	
#Pooling layer	Without pooling	Without pooling	1	Without pooling	Without pooling	Without pooling	Without pooling	
Size of max-pooling	–	–	2	–	–	–	–	
#Dense layers	2	2	2	2	2	2	2	
Activation function of fully-connected	ReLU + sigmoid (last dense layer)	ReLU + sigmoid (last dense layer)	ReLU + sigmoid (last dense layer)	ReLU + sigmoid (last dense layer)	ReLU + sigmoid (last dense la.yer)	ReLU + sigmoid (last dense layer)	ReLU + sigmoid (last dense layer)	
Loss function	Binary cross entropy	Binary cross entropy	Binary cross entropy	Binary cross entropy	Binary cross entropy	Binary cross entropy	Binary cross entropy	
Optimizer, learning rate	Adam,	Adam,	Adam,	Adam,	Adam,	Adam,	Adam,	
0.01	0.01	0.01	0.01	0.01	0.01	0.01	
#Epochs	100	100	100	100	100	100	100	
Batch size	256	256	256	256	256	256	256	
Dropout rate	0.2	0.2	0.2	Without dropout	0.2	0.2	0.2	
Class weight in training?	No	No	No	No	Yes	No	No	

Table 3 The proposed model with different structures.

	DNN-8	DNN-9	DNN-10	DNN-11	DNN-12	DNN-13	DNN-14	DNN-15	DNN-16	DNN-17	
# of convolutional layer	2	3	4	2	4	5	5	4	2	2	
Convolutional layer dim	2D	2D	2D	2D	2D	2D	2D	2D	2D	2D	
# of filters	64, 64	128, 64, 64	128, 128, 64, 64	64, 64	64, 64, 32, 32	256, 128, 64, 32, 16	64, 64, 32, 32, 16	128, 64, 32, 16	32, 32	64, 64	
Kernel size	(3,3), (3,3)	(3,3), (3,3), (3,3)	(3,3), (3,3), (3,3), (3,3)	(3,3), (3,3)	(3,3), (3,3), (3,3), (3,3)	(3,3), (3,3), (3,3), (3,3), (3,3)	(3,3), (3,3), (3,3), (3,3), (3,3)	(3,3), (3,3), (3,3), (3,3)	(3,3), (3,3)	(3,3), (3,3)	
# of Pooling layer	1	1	1	1	1	1	1	1	1	Without pooling	
Size of max-pooling	(2, 2)	(2, 2)	(2, 2)	(2, 2)	(2, 2)	(2, 2)	(2, 2)	(2, 2)	(2, 2)	–	
# of dense layers	2	2	2	2	2	2	2	2	2	2	
Activation function	ReLU + sigmoid (last dense layer)	ReLU + sigmoid (last dense layer)	ReLU + sigmoid (last dense layer)	Tanh + sigmoid (last dense layer)	ReLU + sigmoid (last dense layer)	ReLU + sigmoid (last dense layer)	Tanh + (ReLU & sigmoid: dense layers)	ReLU + sigmoid (last dense layer)	ReLU + sigmoid (last dense layer)	ReLU + sigmoid (last dense layer)	
Loss function	Binary cross entropy	Binary cross entropy	Binary cross entropy	Binary cross entropy	Binary cross entropy	Binary cross entropy	Binary cross entropy	Binary cross entropy	Binary cross entropy	Binary cross entropy	
Optimizer, Learning rate	Adam,	Adam,	Adam,	Adam,	Adam,	Adam,	Adam,	SGD,	Adam,	Adam,	
0.01	0.01	0.01	0.01	0.01	0.01	0.01	0.01	0.01	0.01	
# of epochs	100	100	100	100	100	100	100	100	100	100	
Batch size	1,024	1,024	1,024	1,024	1,024	1,024	1,024	1,024	1,024	1,024	
Dropout rate	Without dropout	Without dropout	Without dropout	Without dropout	Without dropout	0.2 (before dense layers)	Without dropout	Without dropout	0.2 (between dense layesrs)	0.2 (between dense layesrs)	

(1) non-faultyweight=(1/#faulty)∗(total/2.0)

(2) faultyweight=(1/#non-faulty)∗(total/2.0)

In these equations, the term total represents the sum of the instances in the dataset, both faulty and non-faulty. Also, DNN-7 employs two-dimensional convolutional layers to assess the influence of two-dimensional convolutional layers on the proposed DNN performance.

DNN-8 through DNN-17 exhibit a structure akin to DNN-1 through DNN-7, with three notable alterations: 1) Utilizing two-dimensional CNNs to explore the capability of these layers to detect software faults from the adjacent software metrics. This is to leverage the two-dimensional windowing process of CNN layers and extract more spatial information from the element-wise multiplications and summations of feature maps from the adjacent software metrics of the transformed dataset.

2) Transforming the two-dimensional dataset into three-dimensional representation, to include the two new reordered tensors of software metrics into the third dimension of the original dataset. This process also provides more data instances to train the deep learning-based models and potentially enhance the performance of proposed DNN. To accomplish this, we reordered the sequence of numbers from the one-dimensional tensor of the original dataset in two different ways: a) Reversing the original sequence of numbers. For example, if the order of software metrics/features in the original tensor is 0 to 61, the reversed tensor will include the software metrics in the order of 61 to 0.

b) Selecting the software metrics with a specific order: it includes placing the first half of software metrics from the original tensor as the first half of chosen tensor and placing the first half of software metrics from the reversed tensor as the second half of the selected tensor. For example, if the number of software metrics in the original tensor is 61, the chosen tensor takes the first 30 software metrics from the original tensor as its first 30 features (which is the 0 to 30 metrics of the original tensor) and takes the first 31 software metrics from the reversed tensor as its second 31 features (which is the 60 to 31 metrics of the original tensor).

Environment and experiments

To assess the effectiveness of the proposed DNN model, multiple experiments are carried out. Then, a comparison is conducted between the results obtained from this study and those from seven traditional ML models and three state-of-the-art DL classifiers regarding SFP. These experiments are implemented using TensorFlow2.15.0, Keras2.15.0, Python3.10.12, Numpy1.23.5, Pandas1.5.3, Scikit-learn1.2.2, Matplotlib3.7.1 and Imblearn0.10.1. To avoid randomness and providing the reproducibility of the results we utilize random seed of 42 in our experiments. The simulation configuration employed for result extraction comprises an Intel® Core™ i7-6950X CPU operating at 3.0 GHz across ten processor cores, accompanied by 64 GB of RAM. Furthermore, the system utilizes the computational capabilities of an NVIDIA GeForce 1080 GPU to enhance the efficiency of the training phase.

Dataset

The dataset employed in this study is the BugHunter dataset, which is a high-dimensional, automatically created, and freely available bug dataset. This dataset was collected by Ferenc et al. (2020) through the selection of 15 Java projects sourced from the GitHub repository (for full description of the dataset please refer to the Supplemental Materials). It encompasses various granularities such as methods, files, and classes, and contains a broad range of bug information and code metrics. The projects included in this dataset cover the most common software applications, including elasticsearch, orientdb, and neo4j. To assess the dataset’s suitability, the authors applied multiple filters on the raw dataset, employing four methods: GCF, Removal, Subtract, and Single. Among these filtering techniques, Subtract demonstrated superior performance compared to the other methods and authors reported their findings using this filter. Following the authors’ methodology, we present our results utilizing the Subtract filter, intending to compare them with the previous findings. For our experiments, we have used 15 datasets with 74 independent variables/features and one dependent variable/label at the method level. The label of the dataset, ‘Number of Bugs’, contains values starting from 0. A value of 0 signifies that the commit has no bugs, while any other non-zero integer indicates the presence of bugs in the commit. To simplify the classification process, we converted all non-zero values to 1, indicating the presence of bugs in the commit.

Performance evaluation

To assess the effectiveness of the proposed DNN model within SFP, we employed two evaluation metrics: Accuracy and the F1-score, and two time-related metrics: training time (total time required to fit the model using the train data.) and testing time (total time required to generate predictions using the test data). Accuracy measures the closeness of observed values to their true counterparts, and F1-score provides the harmonic average between the precision and recall. Precision quantifies the relevance of the obtained results, while recall measures the accuracy of identified relevant results. These metrics are defined as follows:

Accuracy=TP+TNTP+FP+FN+TN×100, Accuracy=TP+TNTP+FP+FN+TN×100

where: Precision=TPTP+FP×100, and Recall=TPTP+FN×100.

TP (true positive) means the count of predicted faulty methods that are already faulty, TN (true negative) means the count of predicted non-faulty methods that are clean, FP (false positive) means the count of predicted faulty methods that are already clean, and FN (false negative) means the count of predicted clean methods that are faulty.

Performance improvement

This phase aimed to enhance our proposed model performance by fine-tuning four key hyper-parameters: dropout rate, number of epochs, batch size, and learning rate. We evaluated number of epochs from 32 to 500, batch sizes from 32 to 512 in steps of 32, 64, 128, 256 and 512, learning rates from 0.0001 to 0.1, and dropout rates from 0.1 to 0.5. Trials were conducted 15 to 50 times on each project using the Optuna framework in the Python library to tune the hyper-parameters. As the model trained on 15 different projects, the hyper-parameter tuning was done 15 times in total, and each time we measured the performance of the proposed model with accuracy, F1-score, training time, and testing time. Then we compared the performance measures before and after hyper-parameter tuning to assess its effect.

Baseline models

The proposed DNN model is compared with several baseline models. Initially, the traditional ML and boosting baseline classifiers are introduced. Then, the state-of-the-art baseline models are presented. 1) Traditional ML models:

The deployment of traditional ML baseline models was achieved by utilizing the default parameter values provided by the scikit-learn library (Pedregosa et al., 2011). Additional configuration details are outlined in this section.

MLP: A neural network architecture consisting of multiple layers of artificial neurons, renowned for its strong learning capability, resilience to noise, nonlinear processing, parallel processing, fault adaptability, and proficiency in task generalization (Heidari et al., 2020). To ensure network convergence, a maximum of 1,000 iterations was set for all BugHunter projects, except for antlr4, junit, mcmmo, oryx, and titan, which we utilized 2,000 iterations. Moreover, Ceylon-ide-eclipse, mapdb, mct, and neo4j were trained with 1,500 iterations, while Android-Universal-Image-Loader used 3,000 iterations. A random state of one was used for reproducibility of the results. MLP configurations features are the ReLU activation, the Adam solver, the alpha value of 0.0001, constant learning rate, and auto batch size adjustment.

KNN: A non-parametric model that classifies a new data point by determining its class label from the nearest neighbors’ class labels in the training set. This technique is particularly effective for large datasets and low dimensions (Kramer, 2013). The KNN configurations include n neighbors of 5, uniform weights, and auto algorithm.

NB: An algorithm that uses Bayes’ theorem and assumes feature independence to make probabilistic predictions. Naïve Bayes is a frequently used ML model for SFP that yielded efficient results. Bayesian learners were more robust and computationally efficient, but less effective when dealing with correlated features (Pandey, Mishra & Tripathi, 2021). The NB algorithm was trained using the var smoothing of 1e-09.

DT: A tree-based model that recursively splits the data based on feature values to make predictions or classify instances. DT-based models were observed to be more robust, simple, and cost-effective, achieving high accuracy (Pandey, Mishra & Tripathi, 2021). To ensure reproducibility, the random state was set to 42. The DT configurations include gini criterion, max depth 5, and the best splitter.

LR: A linear model used for binary classification, estimating the likelihood of an instance belonging to a specific class based on the input features. LR is easy to implement and very efficient to train (Omri & Sinz, 2020). A maximum iteration of 1,000 and a random state of 42 were utilized for network convergence and reproducibility. The LR configurations include the l2 penalty, intercept scaling of 1, and the lbfgs solver.

RF: An ensemble model that combines multiple DTs to make predictions, using a voting or averaging mechanism to improve the model’s performance. RF classifier is proficient in effectively managing datasets with high dimensionality and multicollinearity, exhibiting speed and robustness against overfitting (Belgiu & Drăguţ, 2016). In this study, a random state of 42 was considered to ensure reproducibility. The RF configurations include n estimator of 100, gini criteria, and sqrt max features.

SVM: A supervised learning algorithm that separates data into distinguished classes by locating a hyperplane that maximizes the margin between the class boundaries. SVM exhibits its capability to tolerate high-dimensional spaces, present robustness to redundant features, effectively manage complex functions, and address non-linear problems (Pandey, Mishra & Tripathi, 2021). The default ‘rbf’ kernel was employed for all of the projects, except for hazelcast and elasticsearch, which used the sigmoid kernel due to the rbf kernel’s inability to predict faulty methods. To facilitate the SVM ability on training the large datasets we utilized batch size of 12,000 for all BugHunter projects, except for hazelcast, orientdb, ceylon-ide-eclipse, and MapDB, which employed batch size of 13,000. Additionally, elasticsearch and netty were trained using batch size of 25,000 and 16,000, respectively. The SVM configurations include the C of 1.0, degree of 3, coef0 of 0.0 and the scale gamma. 2) Boosting algorithms:

Boosting, a popular ensemble learning approach, sequentially trains classifiers using weighted sampling from the initial data and concentrates on challenging instances to improve performance (Song, Guo & Shepperd, 2018). Among the boosting methods, AdaBoost (Freund & Schapire, 1997) stands out as the most widely adopted and influential algorithm in data mining (Wu et al., 2008). Gradient boosting machines (GBM) is another method, which is recognized as a technique to minimize errors and enhance performance progressively (Ayyadevara, 2018). A prominent instantiation of GBM is Extreme Gradient Boosting (XGBoost), renowned for its exceptional performance in supervised learning tasks (Osman et al., 2021). The GBM configurations include the n estimator of 100, learning rate of 0.05, max depth of 3 and the random state of 42.

The XGBoost configurations include the n estimator of 100, learning rate of 0.05, max depth of 3 and the random state of 42.

The AdaBoost configurations include the n estimator of 50, the SAMME algoritm, learning rate of 1.0, and the random state of 42.

3) State-of-the-art models:

The CBIL (Farid et al., 2021) incorporates an embedding layer, a CNN layer, a max pooling layer, a BiLSTM layer, and a dense layer. The CNN and BiLSTM layers utilize the tanh activation, while the dense layer employs the sigmoid activation. CBIL is configured with a dimension of 30 in the embedding layer, 20 filters with a length of 10 in the CNN layer, and 32 LSTM units in the BiLSTM layer.

The 1D-CNN model (Zain et al., 2022) is structured with an Input layer, followed by two one-dimensional convolutional layers, a maxpooling layer, a dropout layer, a Flatten layer, two dense layers, and an output layer. The 1D-CNN model is configured by incorporating 64 filters with a length of 1 in each convolutional layer. Additionally, a pool size of 1 is applied in the maxpooling layer and a rate of 0.3 has been set for dropout layer. The first dense layer utilizes the ReLU activation, while the last dense layer utilizes the sigmoid activation.

The DL-based model (Qiao et al., 2020) comprises an input layer including 20 neurons, two hidden layers including 10 and six neurons respectively, and an output layer including one neuron. The network utilizes the Tanh activation function and a uniform kernel initializer in the first layer. It then continues the learning process with the ReLU activation function in the two subsequent layers, while employing a normal initializer in the last three layers.

Results and discussion

This section will explore the answers to five key research questions (RQs):

RQ1: The impact of employing diverse structures on the DNN performance

Tables 4 and 5 illustrate the performance of 17 different structures for the proposed DNN classifier in terms of F1-score. The best average values are highlighted in bold. Comparing DNN-1 to DNN-7, which utilized the two-dimensional BugHunter data projects, DNN-2, employing two layers of one-dimensional CNNs without the maxpooling layer, 64 filters in each layer, ReLU activation function, Adam optimizer, and a dropout rate of 0.2, demonstrates superior performance in terms of the F1-score. Comparing DNN-8 to DNN-17, which utilized the three-dimensional BugHunter data projects, DNN-17 employing two layers of two-dimensional CNNs without a MaxPooling layer, 64 filters in each layer, ReLU activation function, Adam optimizer, and a dropout of 0.2, demonstrates exceptional performance in terms of the F1-score.

Table 4 The F1-score of the proposed model with different structures.

Name of the project	DNN-1	DNN-2	DNN-3	DNN-4	DNN-5	DNN-6	DNN-7	DNN-2 with 1,024 batch	
ceylon-ide-eclipse	65.10	70.70	71.80	78.82	65.45	73.04	68.72	75.80	
broadleafCommerce	86.09	84.16	84.70	81.43	77.43	81.14	81.54	85.12	
hazelcast	57.48	56.21	52.63	59.80	40.81	56.49	53.64	63.77	
elasticsearch	66.56	76.53	73.91	75.24	51.58	67.73	70.08	70.35	
mapDB	77.20	80.74	81.91	82.99	76.62	84.24	78.90	82.75	
netty	69.32	67.06	68.60	75.89	55.46	64.74	64.53	65.94	
orientdb	72.97	73.40	70.25	75.09	61.65	66.87	67.55	69.97	
neo4j	78.94	80.96	80.57	75.96	62.06	80.28	61.52	77.71	
titan	75.71	82.52	79.59	79.97	80.20	73.32	81.15	81.00	
mcMMO	69.62	69.96	66.54	68.10	69.11	73.35	68.14	64.34	
Android-Universal-Image-Loader	76.10	76.63	78.03	78.89	71.24	79.60	76.39	77.50	
antlr4	80.89	86.50	85.15	83.08	78.95	81.29	80.45	81.61	
junit	75.21	80.03	78.90	83.73	77.63	78.67	75.49	79.73	
mct	96.67	96.67	96.67	74.60	93.33	96.67	96.67	96.67	
oryx	90.65	90.68	90.94	93.20	87.37	93.56	89.80	94.00	
Average	75.90	78.18	77.34	77.78	69.92	76.73	74.30	77.75	

Table 5 The F1-score of the proposed model with different structures.

Name of the project	DNN-8	DNN-9	DNN-10	DNN-11	DNN-11	DNN-12	DNN-13	DNN-14	DNN-15	DNN-16	DNN-17	
ceylon-ide-eclipse	71.19	81.37	77.09	53.04	77.50	78.60	80.62	56.23	60.82	80.29	78.12	
BroadleafCommerce	84.59	82.48	87.05	86.41	83.38	83.24	85.20	81.91	88.45	82.96	85.04	
hazelcast	60.68	68.79	64.56	55.41	62.83	66.28	58.11	64.12	65.48	55.91	62.96	
elasticsearch	75.55	72.09	73.46	65.36	63.59	71.89	75.70	38.90	76.30	77.36	77.57	
MapDB	76.36	76.00	76.20	15.88	78.33	77.42	68.51	76.12	73.69	76.55	81.25	
netty	78.10	75.59	77.30	65.29	72.81	82.94	77.52	77.77	73.65	74.91	78.47	
orientdb	75.92	75.22	73.31	61.49	73.80	78.74	69.30	78.48	68.33	75.06	67.33	
neo4j	76.64	75.20	76.15	66.74	75.40	68.71	72.07	48.20	73.85	78.20	72.44	
titan	80.99	76.81	84.80	84.55	79.52	78.03	79.49	81.87	53.43	81.55	81.97	
mcMMO	59.58	65.07	58.44	48.54	60.29	67.68	69.39	72.94	78.84	67.04	71.83	
Android-Universal-Image-Loader	75.47	61.35	69.60	53.31	69.87	71.01	71.43	43.60	55.59	73.07	70.45	
antlr4	81.87	85.64	85.16	62.56	81.87	82.33	82.98	87.98	43.13	82.10	79.77	
junit	81.61	83.90	81.42	47.27	75.38	82.51	80.24	76.67	58.54	82.14	82.68	
mct	84.21	88.89	89.52	88.89	85.56	74.11	89.52	77.78	66.67	96.67	96.67	
oryx	92.79	88.73	93.51	80.62	91.78	95.36	94.93	92.72	81.60	93.20	92.05	
Average	77.03	77.14	77.83	62.35	75.46	77.25	77.00	70.35	67.89	78.46	78.57	

Since DNN-1 to DNN-7 were trained using the batch size of 256 and DNN-8–DNN-17 utilized a batch size of 1,024, in order to have a fair comparison between one-dimensional and two-dimensional DNNs, we employed a batch size of 1,024 on DNN-2 and present the results at the last column of the Table 4. Considering DNN-5, which included the weights of the imbalanced classes into the learning process of the balanced classes, DNN-1 significantly outperforms DNN-5 in terms of accuracy and F1-score. As a result, including the class weight with the weights of imbalanced classes into training of the DNN using balanced classes could not increase the model performance. When examining DNN-8 to DNN-17, the addition of extra convolutional layers with a significantly higher number of filters, the inclusion of the MaxPooling layer, the absence of the dropout layer, the application of the SGD optimizer, and the adoption of Tanh activation function, along with a combination of Tanh and RelU activations in the model’s architecture, do not significantly influence the prediction of the faulty methods.

Throughout extensive model adjustments, various methods were employed to alleviate overfitting, including regularization, dropout, and class weighting. It is observed that integrating a dropout layer with a rate of 0.2 into the architecture of the proposed DNN has the most significant impact on SFP performance. This outcome can be attributed to the benefits of the dropout mechanism, which aims to reduce model overfitting. In summary, the incorporation of a dropout layer into the DNN model can effectively mitigate overfitting, resulting in a model that is more adept at predicting software faults. Across all the proposed structures, DNN-17 shows superior performance based on the average accuracy and F1-score, suggesting that employing DNN-17 can significantly enhance the accuracy of SFP. In conclusion we consider DNN-17 as the best DNN structure for predicting software faults.

RQ2: Impact of hyper-parameter tuning on the DNN performance

This section discusses the effect of fine-tuning the hyper-parameters on the DNN classifier performance for software fault detection. Table 6 presents the optimal parameters and performance of the DNN classifier after tuning the hyper-parameters. It is observed that the optimal values for each parameter vary across different datasets. On average, tuning the hyper-parameters results in an increase of 6.03% in accuracy and 5.51% in F1-score for the proposed DNN model.

Table 6 The best parameters and performance measures after tuning DNN hyper-parameters.

Dataset	Performance metrics before tuning hyper-parameters	Best parameter	Performance metrics after tuning hyper-parameters	
Acc	F1	Epochs
(32–500)	Batch size
(32–1,024)	Learning rate
(0.0001–0.1)	Dropout rate
(0.0–0.5)	Acc	F1	Pr	Rc	
ceylon-ide-eclipse	69.86	78.12	496	128	0.002	0.3	72.97	81.86	80.70	83.08	
BroadleafCommerce	76.54	85.04	121	32	0.0001	0.5	79.83	87.86	93.22	83.13	
hazelcast	61.89	62.96	358	1,024	0.001	0.2	65.19	70.70	66.50	75.72	
elasticsearch	68.30	77.57	334	1,024	0.009	0.1	69.45	78.50	83.67	73.93	
MapDB	75.34	81.25	381	64	0.0006	0.3	80.82	85.92	87.24	84.65	
netty	70.02	78.47	468	1,024	0.008	0.2	76.78	84.71	82.65	86.96	
orientdb	61.27	67.33	464	1,024	0.006	0.0	74.50	82.29	81.85	82.95	
neo4j	64.44	72.44	343	64	0.0008	0.2	75.39	83.97	87.48	80.83	
titan	72.15	81.97	363	1,024	0.008	0.1	77.22	85.37	85.48	85.66	
mcMMO	61.76	71.83	390	128	0.006	0.1	66.39	74.67	76.15	73.38	
Android-Universal-Image-Loader	60.61	70.45	205	64	0.006	0.4	69.70	77.50	73.91	83.06	
antlr4	69.05	79.77	167	512	0.02	0.1	78.57	86.97	83.78	91.05	
junit	73.40	82.68	212	1,024	0.007	0.4	80.85	87.86	86.84	89.73	
mct	95.45	96.67	76	512	0.0009	0.4	95.45	96.67	93.75	100.00	
oryx	85.80	92.05	478	256	0.01	0.3	93.21	96.39	99.32	93.63	
Average	71.05	78.57		77.08	84.08	84.16	84.51	

Additionally, we investigated the effect of using various kernel sizes on the performance and efficiency of the proposed DNN algorithm. To achieve this, we conducted experiments using kernel sizes ranging from 1 to 5 on 15 datasets. We calculated the average performance metrics for each kernel size and depicted the results in bar graphs (as shown in Fig. 3). Figure 3A indicates that increasing or decreasing the kernel size from three fails to improve the DNN’s accuracy and F1-score in SFP. Figures 3B and 3C show that the increase in kernel size from three fails to enhance the effectiveness of the proposed DNN in SFP.

Figure 3 Effects of various kernel size on DNN performance: (A) Accuracy and F1-score, (B) training time, and (C) testing time.

The receiver operating characteristic (ROC) curve serves as a visual illustration of a model’s ability to discriminate between classes at various decision thresholds. Figure 4 presents the ROC curve and confusion matrix of DNN-17 for the mct, MapDB, and junit datasets. Each pair of sub-figures comprises a plot on the left illustrating the ROC curve and another on the right illustrating the confusion matrix. These figures demonstrate DNN-17’s effective prediction of the faulty methods in the BugHunter dataset.

Figure 4 ROC curve and confusion matrix of DNN-17 on different datasets.

(A and B) mct (C and D) MapDB, and (E and F) junit.

RQ3: The effectiveness of the proposed DNN structure compared to traditional baselines

Table 7 compares the prediction results between the proposed DNN algorithm and seven baseline ML classifiers (MLP, KNN, NB, DT, LR, RF, and SVM) on 15 projects. The suggested DNN algorithm outperforms seven baseline models (MLP, KNN, NB, DT, LR, RF, and SVM) in terms of F1-score, achieving an average value of 84.08%. On average, among seven established machine learning baseline models, RF achieved the highest F1-score of 80.98. Table 8 compares the performance of the proposed DNN algorithm with three boosting-baseline models (GBM, XGBoost, and Adaboost). On average of F1-score, GBM achieved the highest performance among the boosting-based classifiers and DNN surpasses these three algorithms.

Table 7 The F1-score of the proposed model compared to traditional models (part 1).

Name of the project	DNN	MLP	KNN	NB	DT	LR	RF	SVM	Ferenc et al. (2020)	
ceylon-ide-eclipse	81.86	72.51	72.04	3.69	78.71	71.32	83.81	75.79	53.95	
BroadleafCommerce	87.86	85.59	81.13	2.66	77.81	88.85	81.32	88.96	73.66	
hazelcast	70.70	64.52	67.80	76.81	53.34	56.96	72.64	4.73	71.70	
elasticsearch	78.50	72.73	72.02	78.77	71.67	70.31	76.52	2.14	64.11	
MapDB	85.92	76.22	72.56	3.84	75.24	71.20	83.08	70.76	56.10	
netty	84.71	72.87	70.72	79.89	62.17	57.13	80.10	56.31	64.12	
orientdb	82.29	76.67	75.25	81.60	70.39	72.62	80.47	68.46	62.36	
neo4j	83.97	69.30	72.06	81.94	71.93	62.53	83.51	58.45	60.86	
titan	85.37	80.83	75.43	3.17	70.95	75.98	84.85	75.24	62.16	
mcMMO	74.67	68.80	62.92	3.41	73.24	60.90	72.64	67.72	58.15	
Android-Universal-Image-Loader	77.50	74.17	61.55	82.14	87.66	64.91	67.26	71.36	55.69	
antlr4	86.97	82.35	76.35	7.65	74.70	72.60	83.86	67.37	75.73	
junit	87.86	80.57	68.70	13.80	85.70	71.92	81.16	69.75	66.38	
mct	96.67	96.67	83.33	47.97	96.67	65.56	89.52	73.36	68.76	
oryx	96.39	92.87	84.29	46.04	90.96	87.33	94.02	87.99	66.78	
Average	84.08	77.77	73.07	40.89	76.07	70.00	80.98	62.55	64.03	

Table 8 The F1-score of the proposed model compared to traditional models (part 2).

Name of the project	DNN	GBM	XGBoost	Adaboost	
ceylon-ide-eclipse	81.86	88.76	86.76	78.23	
BroadleafCommerce	87.86	82.58	81.15	78.81	
hazelcast	70.70	59.79	56.78	55.33	
elasticsearch	78.50	77.51	78.24	72.36	
MapDB	85.92	85.87	84.83	62.28	
netty	84.71	71.64	66.09	57.29	
orientdb	82.29	85.21	84.20	83.37	
neo4j	83.97	81.71	79.97	67.89	
titan	85.37	87.64	87.48	86.11	
mcMMO	74.67	80.00	79.12	79.04	
Android-Universal-Image-Loader	77.50	84.61	86.25	77.83	
antlr4	86.97	78.01	74.76	76.75	
junit	87.86	89.28	86.15	86.35	
mct	96.67	93.33	96.67	96.67	
oryx	96.39	98.98	98.63	94.58	
Average	84.08	82.99	81.80	76.85	

Additionally, the performance of the proposed DNN, the seven established ML classifiers, and three boosting based classifiers were compared with the best performance presented at Ferenc et al. (2020) on the same datasets. The authors reported their findings using 11 traditional ML classifiers including NB, Naive Bayes Multinomial, OneR, Voted Perceptron, J48 (C4.5), Logistic, SGD, Simple Logistic, Decision Table, RF, and Random Tree employing the Weka library. The last column of Table 7 represents their best findings on 15 projects of the BugHunter dataset. It can be observed that the DNN significantly improve the performance of the fault prediction model on MapDB project by 29.82% (from 56.10 to 85.92) and on antlr4 dataset by 11.24% (from 75.73–considered the best result at the method level, according to the authors findings–to 86.97). Our findings revealed that the proposed DNN algorithm and RF classifier, both utilizing the TensorFlow platform, were able to improve the average F1-score of the traditional classifiers used by Ferenc et al. (2020), which utilized the Weka library, by 20.01% and 16.95%, respectively. Additionally, GBM model enhance the performance of the traditional classifiers used by Ferenc et al. (2020), up to 18.96% on average of F1-score.

RQ4: The effectiveness of the proposed DNN structure compared to state-of-the-art baselines

To address the fourth research question, we conducted a comparison between the performance and efficiency of the proposed DNN classifier and three state-of-the-art DL models: CBIL (Farid et al., 2021), 1D-CNN (Zain et al., 2022), and DL-based model (Qiao et al., 2020). The results are presented in Table 9. In comparison, our proposed DNN exhibits superior performance for predicting faults, in terms of F1-score. Based on our findings, it can be concluded that the proposed DNN model, which incorporates a deep network structure and three-dimensional data representation, exhibits significant potential in detecting the software faults across the 15 BugHunter datasets.

Table 9 The F1-score of proposed model compared to the state-of-the-art models.

Name of the project	DNN	CBIL (Farid et al., 2021)	1D-CNN (Zain et al., 2022)	DL-based (Qiao et al., 2020)	
ceylon-ide-eclipse	81.86	81.81	75.20	80.90	
BroadleafCommerce	87.86	87.45	83.00	74.94	
hazelcast	70.70	77.52	75.09	77.09	
elasticsearch	78.50	79.98	79.13	77.21	
MapDB	85.92	79.99	78.34	78.95	
netty	84.71	82.36	84.18	83.79	
orientdb	82.29	84.03	80.44	82.04	
neo4j	83.97	84.71	80.28	83.11	
titan	85.37	87.39	77.21	83.18	
mcMMO	74.67	78.66	78.98	67.48	
Android-Universal-Image-Loader	77.50	81.14	75.84	80.44	
antlr4	86.97	74.14	86.74	85.01	
junit	87.86	89.22	80.04	86.38	
mct	96.67	84.75	81.83	82.95	
oryx	96.39	93.97	86.07	92.76	
Average	84.08	83.14	80.15	81.08	

RQ5: The most effective SFP classifier based on training and testing times

Figure 5 depicts the training and testing time of various SFP models utilized in this study using 15 BugHunter projects. It is evident that among the range of classifiers analyzed, KNN exhibits the most optimal training time with a value of 0.005, while LR demonstrates the highest efficiency based on the testing time, yielding a value of 0.000.

Figure 5 Training time and testing time of ML and DL models.

Randomness and variability: To address the randomness and account for variability, the experiments were repeated multiple times with different random seeds. The mean, standard deviation, and the confidence interval with confidence level of 95% provided in Table 10.

Table 10 Randomness and variability of the results.

Dataset	Performance metrics with different random seeds	
Accuracy	F1-score	
Mean	Std	Conf. interval
(Conf. Level 95%)	Mean	Std	Conf. interval
(Conf. Level 95%)	
ceylon-ide-eclipse	68.18	6.47	[64.17–72.19]	76.94	7.01	[72.60–81.29]	
BroadleafCommerce	75.57	1.58	[74.60–76.55]	84.45	1.32	[83.63–85.27]	
hazelcast	61.36	1.32	[60.54–62.18]	65.17	4.57	[62.34–68.00]	
elasticsearch	65.24	1.51	[64.31–66.18]	74.02	3.56	[71.82–76.23]	
MapDB	69.21	4.86	[66.20–72.23]	74.44	5.27	[71.17–77.70]	
netty	65.03	5.39	[61.69–68.37]	73.38	5.98	[69.67–77.08]	
orientdb	66.05	3.59	[63.83–68.28]	73.86	4.45	[71.10–76.62]	
neo4j	66.56	4.44	[63.81–69.32]	74.67	5.02	[71.55–77.78]	
titan	70.19	2.99	[68.34–72.04]	79.78	2.64	[78.14–81.42]	
mcMMO	56.68	4.54	[53.87–59.50]	64.09	6.21	[60.24–67.94]	
Android-Universal-Image-Loader	64.70	3.59	[62.47–66.92]	74.23	2.28	[72.81–75.65]	
antlr4	71.25	1.25	[70.48–72.02]	81.68	0.97	[81.08–82.28]	
junit	70.96	2.90	[69.16–72.75]	80.64	2.35	[79.19–82.10]	
mct	93.18	5.47	[89.79–96.57]	94.39	5.79	[90.80–97.98]	
oryx	88.77	1.72	[87.70–89.83]	93.80	1.10	[93.12–94.48]	

Validity of the findings: The validity of the findings pertains to the limitations and potential biases of our results. In this study, we employed experiments using 15 BugHunter data projects at method level. Various fault datasets with different granularity level (class, file, package, method), including NASA MDP, Eclipse dataset, iBUGS, Bugcatchers, and ELFF, are available. As a result, the experimental findings may lack generalizability to alternative datasets and granularity levels, potentially resulting in superior or inferior outcomes for each SFP model of this research. The BugHunter dataset also encompasses a wide range of software metrics. As we mentioned in the Dataset section, the BugHunter dataset comprises source code metrics and clone metrics. Table 4 displays a subset of software metrics included in the BugHunter. Different fault datasets encompass other bug characteristics (static source code metrics, complexity metrics, code smells, and code duplication metrics). It is worth mentioning that different sets of software metrics could lead to varying results. Moreover, through tuning the DNN-17 model, a set of optimal hyper-parameters was obtained for each of the 15 BugHunter datasets, as outlined in Table 7. It is crucial to note that changing the hyper-parameters could lead to differences in the research findings.

Conclusion and future work

This study employed a research approach to investigate the influence of various architectures on the DNN performance within SFP. The primary process of the research method involves constructing DNN classifiers with diverse structures. First, a base structure for the DNN classifier was proposed. Second, an additional 16 DNN models with distinct structures and conditions were built. This is achieved by changing the number of filters in each convolutional layer, the inclusion of the MaxPooling layer, the exclusion of the dropout layer, the inclusion of the weights of the imbalanced classes during training of the balanced classes, the addition of the extra convolutional layers, utilizing different activation functions and optimizers, and employing two-dimensional convolutional layers instead of one-dimensional. We also transformed the dataset into a novel three-dimensional representation to train the two-dimensional convolutional layers in the proposed DNN. Third, we tuned four hyper-parameters (dropout rate, number of epochs, batch size, and learning rate) to enhance the proposed DNN performance. Additionally, we explored the impact of utilizing different kernel sizes on its performance and efficacy. Fourth, we developed seven ML algorithms, three boosting-based techniques, and three state-of-the-art DL models to assess the performance of our proposed model in comparison with these established baselines. The results of these SFP models were analyzed and evaluated in terms of accuracy, F1-score, training time, and testing time.

The key findings of this research is that among 17 proposed DNN structures, seven traditional ML algorithms (MLP, KNN, NB, DT, LR, RF, SVM), three boosting-based techniques (GBM, XGBoost, AdaBoost), and three state-of-the-art DL models, the proposed DNN-17 classifier exhibited superior performance, achieving an F1-score of 84.08% in predicting software faults. Furthermore, this study indicates that including a dropout layer in the structure of the proposed DNN enhances the model performance by minimizing network fluctuations and mitigating overfitting. This improvement has a significant impact on effectively distinguishing between faulty and non-faulty methods. However, making adjustments such as increasing the filter size, considering class weights, adding much more convolutional layers, utilizing Tanh activation function and SGD optimizer do not have a significant effect on software fault detection. Our findings highlight that leveraging two-dimensional convolutional layers with the three-dimensional data representation can significantly increase the SFP performance regarding the F1-score. These findings are highly valuable for software practitioners and researchers in the field.

The findings of this study reveal a considerable improvement in the prediction performance across all 15 projects of the BugHunter dataset. Additionally, we provide optimized values for four hyper-parameters specific to each data project. The results confirm that fine-tuning the hyper-parameters enhances the proposed DNN performance in SFP. In the future, we aim to extend our research by examining the influence of the proposed DNN structure, along with other ML and DL baseline models, at the class or file granularity levels of the BugHunter dataset. Moreover, we can apply additional data preprocessing techniques to improve the prediction performance.

Supplemental Information

Supplemental Information 1 Software metrics of the Bughunter dataset (Ferenc et al., 2020).

Supplemental Information 2 t-SNE visualization of the BugHunter datasets.

We utilized the t-distributed Stochastic Neighbor Embedding (t-SNE) technique to plot the data points of each class.The x-axis and y-axis of each plot represent the two dimensions obtained from reducing the high-dimensional data to two dimensions using t-SNE. Each point in the scatter plot corresponds to an instance of the dataset, and its color is associated with the label of that instance, which is either faulty or non-faulty. The BugHunter dataset comprises source code metrics and clone metrics. From the comprehensive list of software metrics presented at the method level, Table S1 displays a subset of them for simplicity.

Additional Information and Declarations

Competing Interests

Author Contributions

Data Availability

The authors declare that they have no competing interests.

Mehrasa Modanlou Jouybari conceived and designed the experiments, performed the experiments, analyzed the data, performed the computation work, prepared figures and/or tables, authored or reviewed drafts of the article, and approved the final draft.

Alireza Tajary conceived and designed the experiments, analyzed the data, prepared figures and/or tables, authored or reviewed drafts of the article, and approved the final draft.

Mansoor Fateh analyzed the data, authored or reviewed drafts of the article, and approved the final draft.

Vahid Abolghasemi analyzed the data, authored or reviewed drafts of the article, and approved the final draft.

The following information was supplied regarding data availability:

The BugHunter Dataset is available at Mendeley and the University of Szeged:

Ferenc, Rudolf; Gyimesi, Péter; Gyimesi, Gábor; Tóth, Zoltán; Gyimóthy, Tibor (2020), “BugHunter Dataset”, Mendeley Data, V2, doi: 10.17632/8tx7kjbkg4.2.

https://www.inf.u-szeged.hu/~ferenc/papers/BugHunterDataSet/.

Code is available at GitHub:

https://github.com/MehrasaModanlou/DNN-for-Software-Fault-Prediction/blob/main/SFP-DNN.ipynb.

MehrasaModanlou. (2024). MehrasaModanlou/DNN-for-Software-Fault-Prediction: Initial Release (v1.0). Zenodo. https://doi.org/10.5281/zenodo.10719301.

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
