# Peer review of "A novel deep neural network structure for software fault prediction"

_PeerJ Computer Science, doi:10.7717/peerj-cs.2270_

## Round 0.1 · original submission · Major Revisions

The referral process is now complete. While finding your paper interesting and worthy of publication, the referees and I feel that more work could be done before the paper is published. My decision is therefore to provisionally accept your paper subject to major revisions. A detailed comparison is missing.

**Language Note:** PeerJ staff have identified that the English language needs to be improved. When you prepare your next revision, please either (i) have a colleague who is proficient in English and familiar with the subject matter review your manuscript, or (ii) contact a professional editing service to review your manuscript. PeerJ can provide language editing services - you can contact us at [email protected] for pricing (be sure to provide your manuscript number and title). – PeerJ Staff

Reviewer 1 ·

Basic reporting

Overall, the research paper presents an interesting approach to addressing the challenges of software fault prediction using deep learning techniques. However, there are several areas where authors can improve.

Clarify the novelty of the proposed Deep Neural Network (DNN) structure. What specific architectural modifications have been made compared to existing DNN architectures, and why are these modifications important?

Experimental design

Randomness and variability should be addressed in the experimental design. Ensure that experiments are repeated multiple times with different random seeds to account for variability and provide confidence intervals or standard deviations where appropriate.

Validity of the findings

While the results are promising, the paper lacks a comprehensive analysis of the limitations and potential biases of the study. For example, how robust is the proposed DNN model to variations in the dataset or changes in hyperparameters?

Are there specific characteristics of this dataset that may limit the applicability of the proposed approach to other fault prediction tasks or datasets?

Additional comments

The paper would benefit from a more thorough discussion of related work, particularly recent studies that have addressed similar challenges in software fault prediction using machine learning and deep learning techniques.

Cite this review as

Reviewer 2 ·

Basic reporting

Add the confusion matrix and Roc curve of the model you recommend.
You can draw the model you recommend instead of the overall framework.
Which DNN model do you recommend? If DNN-3, highlight this model.
You said you prevented overfitting in contributions and resolved class imbalance. But I couldn't see a different method. Oversampling is a widely used technique. Additionally, the models you create are basic CNN models.

Experimental design

You oversampled. By what percentage did you divide the new data numbers resulting from oversampling? Are there any oversampled samples in the test set?
Almost all of the DNN models you created used ReLU and Adam. Maybe different results could have been obtained by using different activation functions and optimizers. Layer numbers and filter numbers are also close. There is not much difference between the models.
It would have been more comprehensive if you had compared the models you created by creating models such as CNN+LSTM, CNN+LSTM+GRU, CNN+RNN.
Did you keep the learning rate constant or did you use the halving method? Or did you reduce it proportionally?
You said you prevented overfitting, but I don't quite understand how you prevented this. For example, did you add a stopping parameter to the model when you started to overfit? Were the epoch numbers constant?

Validity of the findings

According to Table 7, the model you suggested is inferior to the classical decision tree method. I think the DNN models you create are simple. More successful results can be achieved by creating better models.
Have you applied cross validation technique?
Were oversampling samples used in the test set?
Frankly, it is strange that the decision tree algorithm is better than random forest. You can also compare it with methods such as XGboost, GradientBoosting, Adaboost.
What hyperparameters did you use in classical machine learning classifiers? Briefly talk about it in the article.
Comparison with other studies is important. You only compared it with 3 studies. Are there any other studies done with this data set? If so, add them too.

Cite this review as

---

## Round 0.2 · accepted · Accept

We are happy to inform you that your manuscript has been accepted with minor issues. Since they are related with readability, I think they can be solved in editing process.

Reviewer 1 ·

Basic reporting

Authors updated my previous comments. No further comments.

Experimental design

As above

Validity of the findings

As above

Additional comments

As above

Cite this review as

Reviewer 2 ·

Basic reporting

In addition to the general flow diagram, the figure of the proposed method can also be given in the article so that researchers can better understand your method on the figure.
Roc curve and confusion matrix could be more readable. Resolution and size seem low.

Experimental design

When creating models, I recommend using libraries like Keras Tuner or Gridsearch that help you find the best hyperparameters for hyperparameters like optimizer and activation function selection.

Validity of the findings

Your findings have become stronger with your new experiments and graphs.

Cite this review as